# Prediction on Flow and Thermal Characteristics of Ultrathin Lubricant Film of Hydrodynamic Journal Bearing

**DOI:** 10.3390/mi12101208

**Published:** 2021-10-01

**Authors:** Yulong Jiang, Bo Liang, Zhongwen Huang, Zhenqian Chen, Bo Xu

**Affiliations:** 1School of Energy and Environment, Southeast University, Nanjing 210096, China; jiangyulong@seu.edu.cn; 2AVIC Nanjing Engineering Institute of Aircraft Systems, Nanjing 211102, China; 314312750@seu.edu.cn (B.L.); zw_huang18@126.com (Z.H.); 3Key Laboratory of Energy Thermal Conversion and Control of Ministry of Education, Southeast University, Nanjing 210096, China

**Keywords:** hydrodynamic journal bearing, micro clearance, temperature distribution, flow characteristics, cavitation, heat dissipation

## Abstract

This paper focuses on the flow and thermal characteristics of the lubricant film in the micro clearance of a hydrodynamic journal bearing (HJB) at high rotating speed. A thermohydrodynamic (THD) method consists of the Reynolds equation coupled with energy and viscosity-temperature equation with considering the cavitation is put forward. The 3D surface diagrams of the lubricant film thickness, pressure, temperature, liquid mass fraction, flow rate and heat dissipation distributions under different geometric, operating, slip and no-slip boundary conditions are systemically exhibited and analyzed. The results show that with the rise of eccentricity or length diameter ratio, the maximum peaks of pressure, temperature and heat dissipation are rapidly increased, the cavitation is aggravated, and the flow rate is accelerated in different extent. As the bearing speed accelerating, the maximum peak of temperature is strongly increased, whereas, the distinction between peaks of flow rate and heat dissipation is magnified and reduced, respectively. It provides a fruitful inside view of the inner flow and thermal characterizations of HJB for further understanding its flow-thermal interaction mechanisms and offers theoretical support for improving its working performance.

## 1. Introduction

In accordance with statistics, over 30% of the world’s primary energy is consumed by friction, around 80% of the failure of machine parts are caused by wear, and more than 50% of the fatal accidents related to mechanical equipment are originated from lubrication failure or excessive wear [1]. As an essential type of rotating machinery support element [2], the hydrodynamic journal bearing (HJB) is widely applied in generators [3], turbochargers [4], turbines [5] and vehicle engines [6]. On account of the developing requirement of high efficiency and economy, a dramatic increase of bearing speed has been introduced, which brings about a higher temperature. Meanwhile, as the temperature has effect on the lubricant viscosity, a severer and more complex shear flow occurs in the lubricant film. It is significant to research the flow and thermal characteristics of lubricant film to improve the working stability of HJB. Therefore, many scholars have kept regarding this research topic as area of concern.

For the THD methods, Solghar et al. [7] studied the thermohydrodynamic (THD) characteristics of HJB by employing the SIMPLE algorithm and predicted that the temperature profile can be accurately predicted and used as an assessment of the lubricant feeding conditions. Zhang et al. [8] proposed a THD model of HJB taking mass-conserving cavitation into account, they found that the thermal boundary conditions (TBCs) had great impact on the temperature behavior and supposed the existence of viscous dissipation, whereas, even without further research, the accuracy of the cavitation model was found to be very satisfactory. Feng et al. [9] introduced a misaligned THD model with considering turbulent and claimed that turbulence improves the load capacity while the THD decreases it. In contrast, Mo et al. [10] also adopted the THD method based on dynamic mesh technology, and urged that it can be applied to simulate the variation of inner temperature HJB, whereas, they also mentioned further experimental verification is still of necessity. For the methods of considering cavitation function, Chen et al. [11,12] presented the influence of shape of groove on the lubrication with the cavitation and thermal effect taken into consideration, and provided suggestions for groove shape selection under extreme operation condition, whereas, the viscosity dissipation function was not taken into consideration. Song et al. [13] and Xiang et al [14] introduced the three-dimensional (3D) models with considering conjugate heat transfer and transient interactions for HJB, separately. However, neither the function of heat dissipation nor the slip boundary conditions were not mentioned in these studies. Meanwhile, some scholars adopted the thermoelastohydrodynamic (TEHD) model to research the property of journal bearing. For instance, Thorat et al. [15] computed and measured the metal temperature of bearing by TEHD model and declared that the bearing clearance is important factor influencing the predicted bearing temperature. Nichols et al. [16] investigated the effect of oil feeding rate on bearing performance with a starved flow model based on TEHD. Nevertheless, there is still much room for improvement in the aspects of the versatility and robustness of TEHD models. Besides, some scholars choose to study HJB through CFD based modeling [17,18,19,20]. For instance, Ding et al. [17] analyzed the effect of air on oil-air distribution and energy characteristics. Armentrout et al. [18] developed a method to adjust the turbulence model within the Reynolds solution. Hagemann et al. [19] introduced a model to describe the influences of leading edge grooves on operating characteristics. Li et al. [20] reported a new type of dynamic mesh algorithm based on s structured grid. However, the function of heat dissipation and slip boundary conditions were not given sufficient attention to, either. In contrast to the numerical methods, some scholars also choose experiment as their investigation tools. Liu et al. [21] experimentally studied the lubrication of water-lubricated HJB which can be a support for the simulation results. Zhang et al. [22] explored the mechanism of synergetic lubrication by both experiment and CFD, and emphasized the significant influence of lubricant viscosity. Other scholars also pay attention to the investigation of journal bearings [23,24,25,26]. Li et al. [23] investigated the microbearings and found that the rise of temperature would weaken the load capacity and negatively influence the operating stability of microbearings. Wu et al. [24] studied the static and dynamic characteristics of gas bearings based on viscosity model of Veijola and declaimed that the effective viscosity of the lubricant would lead to a decreased load carrying capacity. Besides, Chen et al. [25] focused on the fabrication of a biomimetic superhydrophobic surface in the application of a water-lubricated bearing, which enlarged the research scope of surface lubrication of bearings. Yu et al. [26] developed new applications of silica film in micro air bearings, which broadened the manufacture craft of bearings. However, the interactions between the viscosity and temperature of lubricant should be further taken into consideration. Besides, the influence of geometric and operational parameters on the lubricant film thickness, pressure, temperature, liquid mass fraction, flow rate and heat dissipation distributions need to be exhibited and analyzed systematically and adequately.

In this study, we mainly focus on the flow and thermal characteristics of the ultrathin lubricant film in the micro high-speed rotating clearance between the journal and shell surfaces of HJB. The numerical model coupled the simultaneous solution of Reynolds equation, energy equation, and Roelands viscosity-temperature equation with considering the functions of cavitation and the relationship between viscosity and temperature is established. The influences of eccentricity, length diameter, clearance ratios and bearing speed on the lubricant film thickness, pressure, temperature, liquid mass fraction, flow rate and heat dissipation are systemically studied, exhibited and discussed. Besides, the effects of the no-slip and slip boundary conditions on the flow and thermal properties of the ultrathin lubricant film of HJB are compared and analyzed. The Sommerfeld number is also introduced for presenting the results more synthetically. Through this investigation, we aim to provide fruitful new insights into the flow and thermal characteristics of HJB, exhibiting the interesting 3D surface diagrams of lubricants in the tiny clearance, and supply an efficient and effective tool to improve the design and promote the lubrication and working performance of HJB in engineering design and applications.

## 2. Numerical Method

### 2.1. Physical Model

The structure of HJB is displayed in Figure 1. The lubricant film filling in the clearance between the surfaces of journal and bearing house is marked. When the journal starts to rotate in the anticlockwise direction, the lubricant continuously accumulates in the convergent wedge, leading to the increase of pressure adjoining the minimum lubricant thickness position. After the journal speed reaching a certain value, the position of journal is shifted to right, the pressure resultant force generated by lubricant film is balanced with load *W*.

### 2.2. Governing Equations

#### 2.2.1. Reynolds Equation

The generalized Reynolds equation [27] with considering the variation of viscosity throughout the lubrication film expressed in cylindrical coordinate can be written in dimensionless form as follows:(1)$$\frac{\partial }{\partial \theta }\left(\frac{H^{3}}{\eta^{*}}\frac{\partial P}{\partial \theta }\right)+\alpha^{2}H^{3}\frac{\partial }{\partial Y}\left(\frac{H^{3}}{\eta^{*}}\frac{\partial P}{\partial Y}\right)=\frac{dH}{d\theta }$$
(2)$$H=\frac{h}{C}=1+\varepsilon \cos\theta$$
where $Y=\frac{y}{b}$; $P=\frac{pc^{2}}{6U\eta_{0}R}$; $\theta =\frac{x}{R}$; U=Rω. *Y* is dimensionless axial coordinate; *b* is bearing width; *c* is radial clearance; *θ* is angular coordinate; *ε* is eccentricity ratio.

#### 2.2.2. Energy Equation

In lubrication calculation, the generalized energy equation [28] can be expressed as the following form:(3)$$\rho c_{p}\left(u\frac{\partial T}{\partial x}+v\frac{\partial T}{\partial y}+w\frac{\partial T}{\partial z}\right)=k\left(\frac{\partial^{2}T}{\partial x^{2}}+\frac{\partial^{2}T}{\partial y^{2}}+\frac{\partial^{2}T}{\partial z^{2}}\right)-\frac{T}{\rho }\frac{\partial \rho }{\partial T}\left(u\frac{\partial p}{\partial x}+v\frac{\partial p}{\partial y}+w\frac{\partial p}{\partial z}\right)+\Phi$$
where *c_p_* is specific heat capacity at constant pressure; *u*, *v*, *w* represent velocity components; *k* is heat transfer coefficient; *η* is viscosity; *Φ* is viscosity dissipation [28], which can be simplified expressed as:(4)$$\Phi =\eta \left[{\left(\frac{\partial u}{\partial z}\right)}^{2}+{\left(\frac{\partial v}{\partial z}\right)}^{2}\right]$$

After dimensionless normalization, the energy equation in cylindrical coordinate is expressed as:(5)$$\frac{\partial T^{*}}{\partial \theta }=\frac{1}{Q_{x}}\left\{-\alpha Q_{y}\frac{\partial T^{*}}{\partial Y}+\frac{2\eta^{*}}{H}+\frac{6H}{\eta^{*}}\left[{\left(\frac{\partial p}{\partial \theta }\right)}^{2}+\alpha^{2}{\left(\frac{\partial p}{\partial Y}\right)}^{2}\right]\right\}$$
(6)$$Q_{x}=\frac{H}{2}-\frac{H^{3}}{2}\frac{\partial P}{\partial \theta };\;Q_{y}=-\frac{H^{3}}{2}\frac{\partial P}{\partial Y}$$
where η_0_ is inlet viscosity, *η** is dimensionless viscosity, $\eta^{*}=\frac{\eta }{\eta_{0}}$; $T^{*}=\frac{2J\rho c_{p}h_{0}^{2}}{UR\eta_{0}}T$.

#### 2.2.3. Viscosity-Temperature Equation

Rolelands Equation in dimensionless form [29]:(7)$$\eta^{*}=exp\left\{\left(\ln\eta_{0}+9.67\right)\left[{\left(\frac{T-138}{T_{0}-138}\right)}^{-1.1}-1\right]\right\}$$

Density-temperature Equation
(8)$$\rho =\rho_{0}\left[1-\alpha_{T}\left(T-T_{0}\right)\right]$$
where, ρ is lubricant density under temperature T, $\rho_{0}$ is lubricant density under temperature $T_{0}$, $\alpha_{T}$ is thermal expansion coefficient, ${℃}^{-1}$, which can be calculated as follows:(9)$$\alpha_{T}=\left\{\begin{array}{c} \left(10-\frac{9}{5}\mathrm{lg}\eta \right)\times 10^{-4}\;\mathrm{lg}\eta \leq 3.5\;\\ \left(5-\frac{3}{8}\mathrm{lg}\eta \right)\times 10^{-4}\;\mathrm{lg}\eta >3.5\; \end{array}\right.$$

#### 2.2.4. Cavitation

The analysis of cavitation is on the basis of a modified version of the Elrod algorithm [30,31]:(10)$$v_{av}=v_{av,\;c}-gv_{av,\;p}\nabla_{t}p_{f}$$
where *v_av,c_* and *v_av,p_* represent to the average Couette velocities and the average Poiseuille velocities, respectively. The density of lubricant is set as follows:(11)$$\rho =\rho_{c}e^{\beta p_{f}}$$
where *β* is the compressibility, *ρ_c_* is the density at cavitation pressure. The variable *ξ* corresponding to the fractional film content, when *ξ* < 1 refers to the cavitation region, is defined as follows:(12)$$\xi =\frac{\rho }{\rho_{c}}$$

#### 2.2.5. Sommerfeld Number

The Sommerfeld number *S* can be calculated as follows [32]:(13)$$S=\frac{\mu \omega LD}{W}{\left(\frac{R}{C}\right)}^{2}$$
where *μ* is Dynamic viscosity, Pa·s^−1^, *ω* is bearing speed, rpm, *W* is bearing load, N.

### 2.3. Boundary Conditions

#### 2.3.1. No-Slip Boundary Conditions

The no-slip boundary conditions of the dimensionless pressure and temperature are as follows:

In axial direction:(14)$${\left.P\right|}_{Y=\pm 1/2}=0;\;T_{1,j}^{k+1}=\frac{T_{1,j}^{k}+T_{N,j}^{k}}{2}$$

In circumferential direction:(15)$${\left.P\right|}_{\theta =0}=0;\;{\left.P\right|}_{\theta =\theta_{2}}=0;\;{\left.\frac{\partial P}{\partial \theta }\right|}_{\theta =\theta_{2}}=0;{\left.\frac{\partial T}{\partial Y}\right|}_{Y=0}=0$$

The temperature calculation grid is exhibited as Figure 2. The flow chart of the calculation iterative process is exhibited in Figure 3. The relevant geometric and operating parameters can be seen in Table 1.

#### 2.3.2. Slip Boundary Conditions

Assuming a slip length of *L_sj_* at the journal, and a slip length of *L_sb_* at the bearing, the general slip boundary conditions can be expressed as:(16)$$v_{av}=v_{av,c}-v_{av,p}\nabla_{t}p_{f}$$
where $v_{av}$ is a term related to Couette flow, and $v_{av,p}$ is a coefficient concerned with Poiseuille flow.

The force terms can be expressed as follows:(17)$$f_{j}=-f_{j,p}\nabla_{t}p_{f}+f_{j,c}$$
(18)$$f_{b}=-f_{b,p}\nabla_{t}p_{f}+f_{b,c}$$
where $f_{j,p}$ and $f_{b,p}$ are the Poiseuille coefficient for the force on the journal and bearing, respectively; and $f_{j,c}$ and $f_{b,c}$ incorporate the Couette and normal forces (on account of the pressure) on the journal and bearing, respectively.

### 2.4. Mesh Independence Verification

In order to verify the independence of grid partition, we gradually intensify the gridding of the model from 1.06 × 10^3^ to 4.97 × 10^4^ and the variation of maximum dimensionless pressure with number of grids is shown in Figure 4, of which, the relevant calculation parameters are displayer in Table 2. It is shown that the pressure keeps basically constant after the number of grids exceeds 1.57 × 10^4^ and the mean deviation of the numerical result is less than 0.9%. Comprehensively considering the computing time and precision factors, we choose it as the grid independent solution.

### 2.5. Validation of Model

As is shown in Figure 5, the comparison between the maximum temperature and pressure versus eccentricity ratio from the experiment data in literature [33] and numerical results simulated by model in this paper at *ω* = 2 × 10^3^ rpm and *ω* = 4 × 10^3^ rpm, respectively, the relevant calculation parameters are shown in Table 3. For the maximum temperature displayed in Figure 5a, the predictions of the model are in good agreement with experiment results. In addition, the maximum pressure is shown in Figure 5b, it can be seen that the tendency of the predicted results with considering cavitation function are in better consistence with experimental data than those without considering. Therefore, it can be concluded that the THD model containing the interactions between the lubricant viscosity and temperature and the function of cavitation can well exhibit the flow and thermal characteristics of lubricant film in HJB.

## 3. Results and Discussion

### 3.1. Effects of Eccentricity Ratios

In this section, the eccentricity ratio *ε* will rise from 0.3, 0.5 to 0.7, with other parameters set as *C*_0_ = 180 μm, *ω* = 3 × 10^3^ rpm, *L*/*D* = 0.83, α = 0.003. It is depicted in Figure 6 that with the eccentricity ratios ε increase from 0.3, 0.5 to 0.7, the dimensionless maximum film thickness elevates from 1.3, 1.5 to 1.7, with a growth rate of 30.8 %, and the dimensionless minimum film thickness descends from 0.7, 0.5 to 0.3, with a reduction rate of −57.1 %. On account of higher eccentricity ratio, the load on bearing shaft is aggravated, and the acting force on the lubricant is strengthened. As a consequence, the minimum clearance of the bearing becomes narrower in size and the dimensionless minimum lubricant film thickness diminishes. In contrast, the maximum clearance of the bearing becomes larger and the dimensionless maximum lubricant film thickness broadens.

The variations of dimensionless maximum lubricant pressure with different eccentricity ratios is displayed in Figure 7. With the eccentricity ratios *ε* ranging from 0.3, 0.5 to 0.7, the dimensionless maximum lubricant pressure increases from 0.16, 0.48 to 1.57, with different increase rates as much as 200 % and 227 % at the two intervals between *ε* = 0.3/0.5, and *ε* = 0.5/0.7. It is observed that the dimensionless maximum pressure rises more rapidly at higher eccentricity ratio. It can be caused by the higher eccentricity ratio is, representing the larger bearing load compression function on the bearing shaft is and the stronger squeezing action on the minimum thickness position will be, as a result, a higher maximum pressure of the lubricant film will get.

It is exhibited in Figure 8 that the lubricant temperature distributions change with the eccentricity ratios *ε* rise from 0.3, 0.5 to 0.7. The dimensionless maximum lubricant temperature difference increases obviously from 0.42 °C, 1.03 °C to 6.41 °C, with growth rates of 145.2% and 522.3% at the two lifting stages of eccentricity ratios. The rise of maximum temperature in the latter stage is as much as over 8.8 times that of the former stage. In addition, at *ε* = 0.3, the temperature peak is not evident. When it reaches *ε* = 0.5, the temperature peak emerges. As it arrives at *ε* = 0.7, three temperature peaks are apparent accompanying with the rapid promotion of temperature. Whereas, the Sommerfeld number declines quickly at the beginning then slowly with the increase of eccentricity ratio. It is due to that the Sommerfeld number is inversely proportional to the reciprocal of bearing load. As the eccentricity ratio can positively signify the quantity of bearing load, that the relationship between the Sommerfeld number and the eccentricity ratio is reasonably.

It can be explained as, at higher eccentricity ratio, stronger compressible function makes minimum lubricant film thinner, discharge of the large amount of energy produced by intensified shearing action at high rotating speed will be more difficult, and resulting the fast increase of temperature. Furthermore, the profile of temperature distribution is not as smooth and steady as those of film thickness and pressure, which is caused by the comprehensive effects of the temperature gradients in the radial and the axial directions and the variations of film thickness. Li et al. [34] reported that the temperature distribution spreads in a strip, on the journal interface, whereas in a square, on the bearing interface, nevertheless, they did not mention the temperature distribution in the lubricant film itself, and the results in this research can be regarded as a supplement and an explanation for the temperature distributions on the surfaces.

The comparison of the lubricant liquid mass fraction, flow rate and heat dissipation are displayed in Figure 9, Figure 10 and Figure 11, respectively. With the eccentricity ratios *ε* adding up from 0.3 to 0.7, the minimum liquid mass fraction drops rapidly up to −53.4%, which indicates more severe cavitation function occurs. Meanwhile, the maximum flow rate, which appears around the minimum wedge-shaped clearance, increases steadily as 8.9%, in contrast, the minimum flow rate, which lies in the maximum thickness clearance declines more obviously, as high as, −31.1%, shows larger rate differences generate as the eccentricity ratios increase. Besides, the maximum heat dissipation first markedly improves then slowly increases up as high as 155.2 times, whereas, the minimum heat dissipation distinctly decreases as much as −41.3%, which indicates more heat dissipation will be produced as the bearing load on the journal becoming larger. The friction heat caused by the stronger heat dissipation at higher eccentricity ratio should be emphasized. In addition, under the investigation conditions, the effect of slip boundary on the properties does not show distinct difference from that of the no-slip boundary.

### 3.2. Effects of Length Diameter Ratios

In this section, the length diameter ratios *L*/*D* will increase from 0.5, 0.83 to 1.0, while other parameters keep as *C*_0_ = 180 μm, *ω* = 3 × 10^3^ rpm, *ε* = 0.7, *α* = 0.003. As is displayed in Figure 12, the dimensionless lubricant film pressure distributions change with different length diameter ratios. When *L*/*D* increases from 0.5, 0.83 to 1.0, the dimensionless maximum lubricant temperature lift of lubricant film elevates from 3.92, 6.41 to 7.55, with a growth rate at 92.6%. The variation tendency of the simulation results is consistent with the experimental temperature distributions at similar bearing speed by Ftizgerald et al. [35], which is a validation for the method in this paper. At a larger length diameter ratio, the dimensionless mass flow rate will be enhanced and more flow mixing effect will appear in the bearing clearance, resulting in more viscous heat produced, and leading to the temperature increased rapidly. As shown in Figure 12d, the maximum lubricant temperature increases almost linearly with the growth of length diameter ratio of HJB, which is a verification of the great impact by dimensionless mass flow rate, friction and viscous heat on the lubricant film temperature distribution. Besides, with the increase of *L*/*D*, the Sommerfeld number grows stably at a rate of 99.6%. Because the Sommerfeld number is positively proportional to the bearing length. It indicates that the bearing capacity is strengthened steadily with the rise of *L*/*D*.

The influence of *L*/*D* on the dimensionless lubricant film pressure distributions is shown in Figure 13. With the *L*/*D* increasing, the dimensionless maximum lubricant pressure of lubricant film elevates at a rate of 97.3 % which shows the friction moment and the bearing capacity are improved at the same time. It is exhibited in Figure 14, Figure 15 and Figure 16 that the lubricant liquid mass fraction, flow rate and heat dissipation vary with different bearing length diameter ratios *L*/*D*. With *L*/*D* adding from 0.5 to 1.0, the minimum liquid mass fraction declines as much as −41.0%, indicating cavitation is intensified. Meanwhile, the maximum flow rate, increases steadily as 7.7%, in contrast, the minimum flow rate, which lies in the maximum thickness clearance decreases rapidly, up to −31.8%. It manifests that at higher *L*/*D* the rate difference will be larger than that of lower *L*/*D*. The maximum heat dissipation rises up to 902%, in contrast, the minimum heat dissipation slightly changes as much as 0.79%, which indicates the influence of *L*/*D* on the maximum heat dissipation is far severe than that of the minimum heat dissipation, declaring a higher temperature difference exiting. Moreover, comparing the results under different boundary conditions, it can be easily found that the influence of slip boundary on heat dissipation is more remarkable than those of other characteristics. For minimum heat dissipation, the values of slip conditions are always lower than those of no-slip conditions. In contrast, for the maximum heat dissipation, the value of slip boundary at high *L*/*D* surpasses that of no-slip, which manifests that too much long bearing length may not be beneficial to the thermal stability of HJB. The results in this section can be regarded as a reference in choosing the appropriate boundary dimensions of HJBs.

### 3.3. Effects of Bearing Speeds

In this section, the bearing speeds *ω* will add up from 3 × 10^3^ rpm, 6 × 10^3^ rpm to 9 × 10^3^ rpm, with other parameters set as *C*_0_ = 180 μm, *ε* = 0.7, *L*/*D* = 0.83, *α* = 0.003. Seen from Figure 17, the temperature difference distributions transform at different bearing speeds. When the bearing speed accelerating, the maximum temperature difference of lubricant increases from 6.41 °C, 12.9 °C to 19.5 °C, respectively, at a growth rate of 204.2% in total. Besides, three temperature peaks appear in each figure and all of them lift with the growth of bearing speed, which is caused by the comprehensive effects of the temperature gradients in radial and axial directions and the variations of film thickness. To be more specific, the temperature of lubricant film is inversely proportional to the lubricant film thickness and is proportional to the quadratic of the velocity gradients in both the axial direction and the radial direction. Meanwhile, the Sommerfeld number is rapidly increased at a growth rate of 200% in total with the acceleration of bearing speed from 3 × 10^3^ rpm to 9 × 10^3^ rpm, which is caused by the directly proportional relationship between the Sommerfeld number and the bearing speed.

With the bearing speed increases, the viscosity dissipation, showing in Equation (4), is intensified and the aerodynamic friction heat is heavily accumulated and the maximum lubricant temperature boosts up. Besides, the maximum temperature enhancement of lubricant film is linearly with the increase of the bearing speed, which can be explained as follows. On one hand, with the bearing speed accelerating, the viscous shear action of lubricant film is intensified and the friction heat is accumulated and the temperature of lubricant film is increased. On the other hand, with the temperature of lubricant increasing, the viscosity of lubricant will be decreased and the viscosity dissipation heat will be reduced in certain extend. While the relationship between the maximum temperature enhancement difference of lubricant film and the increase of the bearing speed is linear, a dynamic balance might exit between the variations of bearing speed and the viscosity of lubricant viscosity.

The comparison of pressure, liquid mass fraction distributions under different bearing speeds are depicted in from Figure 18 to Figure 19, respectively. The maximum pressure increases as high as 170.6%. On the contrary, interestingly, the minimum lubricant liquid mass fraction nearly unchanged, whereas the area of the cavitation region expands. As reported by Poddar et al. [36], the photographs study indicated that at different bearing speed the discrete vapor bubbles are greatly different in size inside the cavitation region, and with the bearing speed adding up, a relative decline in the size of bubbles, meanwhile, a growth in vapor bubbles number being evolved, resulting the minimum mass fraction keeps the same, whereas, the cavitation area becomes larger. Similarly, the distribution of volume fraction of the vapor phase has been shown by Li et al. [36], in the shape of square, comparing with which, the simulated figures in this study probably better consistent with the experiment results. Besides, Dhande et al. [37] also researched the flow phenomena through 3D CFD analysis and gave some figures of vapor volume fraction of oil vapor from 1 × 10^3^ rpm to 5 × 10^3^ rpm, which are similar to the results in this paper. In one hand, the experimental results in the research of Dhande et al. [37] can be a validation for the model in this study. On the other hand, the bearing speed in this study arrives at as high as 9 × 10^3^ rpm, which can be considered as a break through and transcending of the researches of predecessors.

In addition, as is shown in Figure 20 and Figure 21 the maximum flow rate increases promptly up to 202.7%, whereas, the minimum flow rate lifts as high as 245. %, however, the absolute value of the latter is as 65.6% as that of the former. In contrast, the maximum heat dissipation unexpectedly declines as much as –33.5%, on the contrary, the minimum heat dissipation distinctly increases up to 826.5%, which indicates difference between the maximum and minimum heat dissipation shrinks with the bearing speed accelerates. Moreover, the influence of slip boundary on the heat dissipation distribution is more obvious than those of other properties. Interestingly, at high bearing speed at 9 × 10^3^ rpm the maximum heat dissipation under slip boundary is higher than that of no-slip boundary, which is not as expected. It can be due to the slip function at high bearing speed is strengthened and more shear driving force is consumed for dissipation heat production instead of active work on the journal.

### 3.4. Effects of Clearance Ratios

In this section, the clearance ratio α will increase from 0.002, 0.003 to 0.004, whereas other parameters are kept as *C*_0_ = 180 μm, *ω* = 3 × 10^3^ rpm, *ε* = 0.7, *L*/*D* = 0.83. It is depicted in Figure 22 that with the increase of clearance ratios α, the dimensionless maximum temperature difference of lubricant film declines from 14.6 °C, 6.41 °C to 3.59 °C, respectively, with a reduction rate of −75.4% in all. The maximum temperature of lubricant film decreases rapidly with the growth of the clearance ratio.

It can be explained as with the increase of the clearance ratio, the clearance of the minimum thickness is enhanced and the flow channel of lubricant is larger than before, leading to decline of the mass flow rate and the decrease of friction heat of the lubricant film at the same time. The tendency of maximum temperatures is in well accordance with the results by Solghar et al. [38]. However, they gave the results as the temperature versus different clearance ratios in a 2D diagram. The 3D figures in this research can provide the overall temperature distribution information in a more intuitively and comprehensively manner, which can be regarded as an improvement on the basis of the previous studies. In addition, as the clearance ratio is enlarged, the Sommerfeld number is decreased at a fast rate of −75% in all, which can be led by the inversely proportional relationship between the Sommerfeld number and the bearing clearance ratio.

As is displayed in Figure 23, Figure 24, Figure 25 and Figure 26 that the lubricant film pressure, liquid mass fraction, flow rate and heat dissipation vary with different clearance ratios *α*. With α adding from 0.002 to 0.004, the maximum pressure decreases rapidly at −72.4%, in contrast, the minimum liquid mass fraction slight rises as 15.0%. On the other hand, the maximum flow rate decreases scarcely as much as −0.91%, meanwhile, the minimum flow rate decreases more obviously, up to −18.4%. In contrast, the variation of the maximum heat dissipation is much more severely, as high as 261.1%, on the contrary the minimum heat dissipation declines up to −75.5%. It indicates that the clearance ratios have greater effect of on the maximum heat dissipation than the minimum heat dissipation. In addition, the influences of slip and no-slip boundaries on the flow rate and heat dissipation are more evident than those of other factors. For the former, as the clearance ratio magnifies, the minimum flow rate gradually of slip-boundary becomes less than that of no-slip boundary, which indicates that too much larger clearance is not good for the bearing working performance. For the latter, with the clearance ratio increasing, the maximum heat dissipation under slip condition slightly changes from higher to lower than those under no-slip condition, which can be caused by in the larger space under greater clearance ratio the slip function is weakened in some extent and the dissipation heat is relatively more easily to be discharged.

## 4. Conclusions

The flow and thermal characteristics of the lubricant in the micro high-speed rotating clearance between the journal and shell surfaces of HJB is important issue, which have great effect on HJB working stability. In this research, the HJB is numerically simulated by coupling the dimensionless Reynolds equation, energy equation and Roelands viscosity-temperature equation with considering cavitation function and the interactions between the lubricants viscosity and lubricants temperature. The 3D surface diagrams of the lubricant film thickness, pressure and temperature distribution under different eccentricity ratios, length diameter ratios, bearing speeds and clearance ratios are displayed and discussed.

An increase in eccentricity ratio enlarges the differences between maximum and minimum peaks of thickness, flow rate and heat dissipation, severely strengthens the maximum peaks of pressure and temperature, and aggravates the degree of cavitation.A growth in length diameter ratio leads to the rise of the maximum peaks in pressure and temperature distributions, and promotes the degree of cavitation, meanwhile, intensify the differences between the maximum and minimum peaks of flow rate and heat dissipation.A rise in the bearing speed strongly increases the maximum peak of temperature, magnifies the distinctions between maximum and minimum peaks of flow rate, nevertheless, shrinks those of heat dissipation, and has slightly influence of cavitation.A lift in clearance ratios result in a sharply decline in maximum peak of temperature, exaggerating the differences between maximum and minimum peaks of heat dissipation, a mild decrease in the minimum flow rate, and slight effect on the maximum flow rate and cavitation.It is suggested that the optimization conditions for HJB can be a relative high eccentricity, medium length diameter ratio, relative high bearing speed, as well as medium clearance ratio, and in this research the combination could be *ε* = 0.6, *L*/*D* = 0.7, *ω* = 6 × 10^3^ rpm and *α* = 0.003.

We provide new insights into the flow and thermal characteristics of HJB, exhibiting the interesting 3D surface diagrams of lubricants in the tiny clearance, and supply an effective tool to research and promote the lubrication and working performance of HJB in engineering applications. Future work will focus on experimental investigation on flow, pressure and temperature distributions (by processing bearing test rigs with different geometric parameters and surface roughness, etc.), and optimizing the numerical model with experimental results, therefore, improving its accuracy and practical value.

## Figures and Tables

**Figure 1 micromachines-12-01208-f001:**
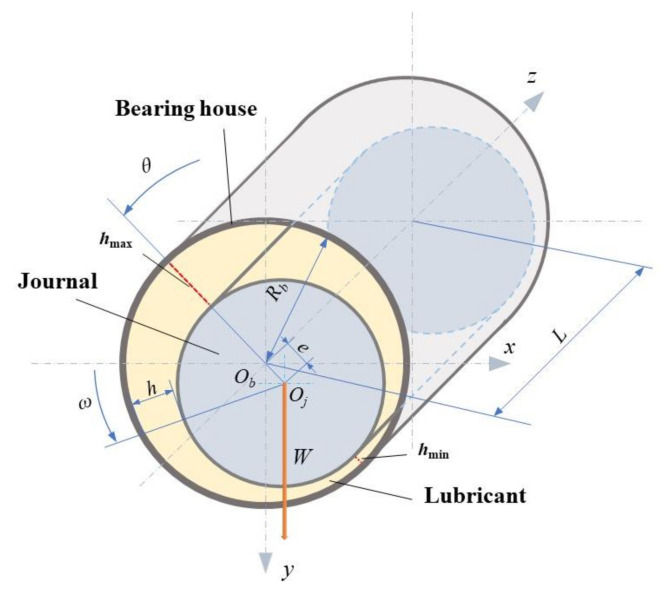
Schematic diagram of HJB structure.

**Figure 2 micromachines-12-01208-f002:**
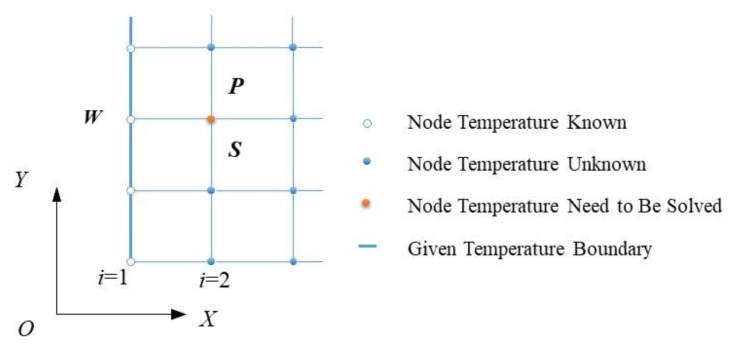
Temperature computational grid and boundary conditions.

**Figure 3 micromachines-12-01208-f003:**
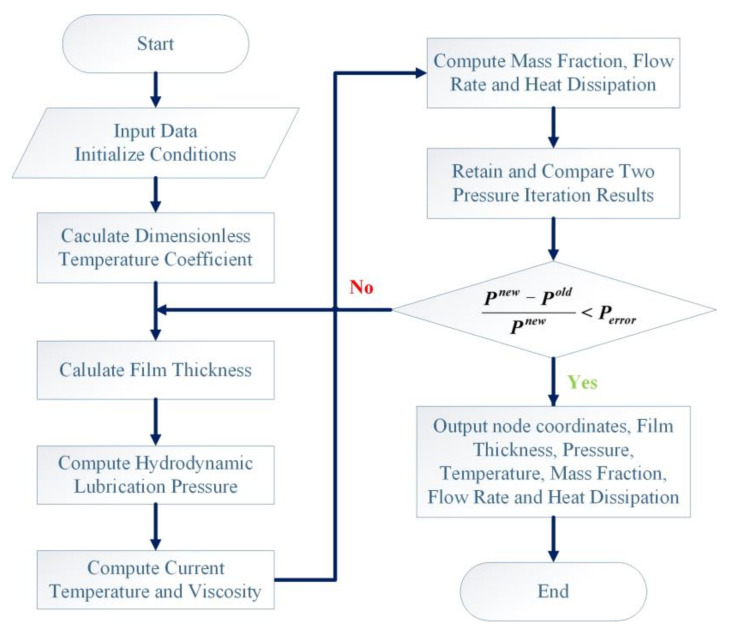
Calculation iterative process.

**Figure 4 micromachines-12-01208-f004:**
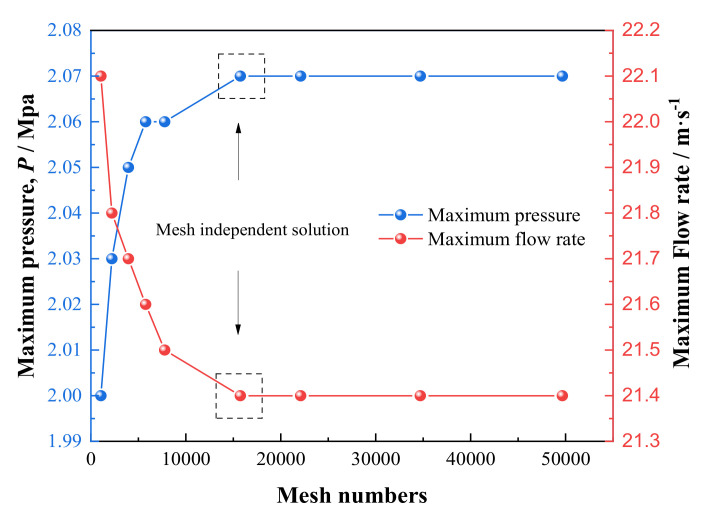
Mesh independence verification.

**Figure 5 micromachines-12-01208-f005:**
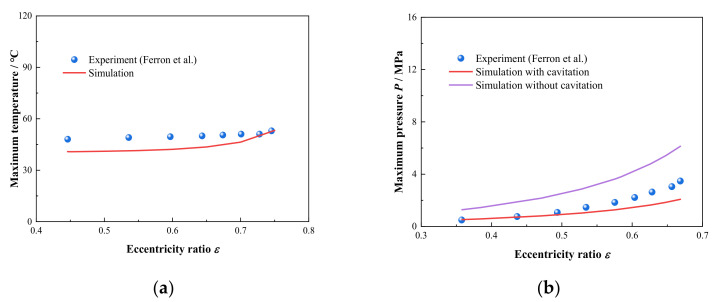
Comparison of simulation results with experiment data in literature (**a**) Maximum temperature; (**b**) Maximum pressure.

**Figure 6 micromachines-12-01208-f006:**
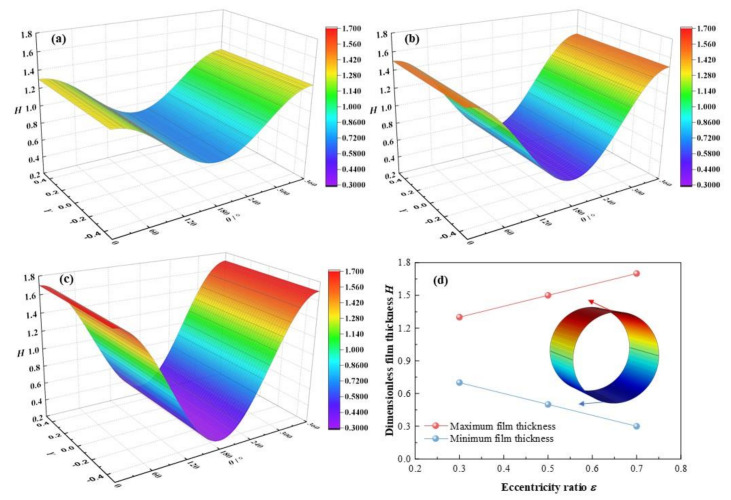
Comparison of the film thickness distribution under different eccentricity ratios. (**a**) *ε* = 0.3; (**b**) *ε* = 0.5; (**c**) *ε* = 0.7; (**d**) maximum/minimum film thickness.

**Figure 7 micromachines-12-01208-f007:**
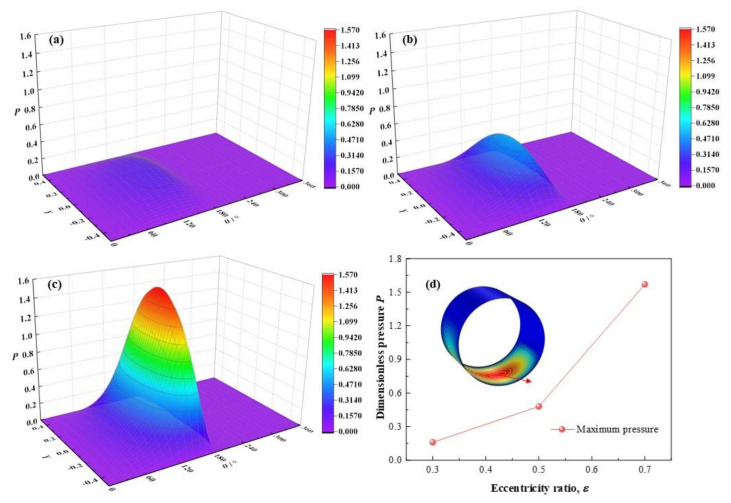
Comparison of the pressure distribution under different eccentricity ratios. (**a**) *ε* = 0.3; (**b**) *ε* = 0.5; (**c**) *ε* = 0.7; (**d**) maximum pressure.

**Figure 8 micromachines-12-01208-f008:**
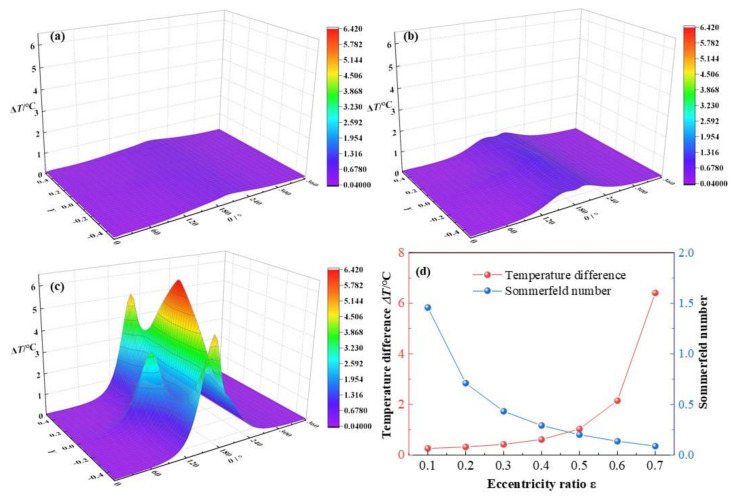
Comparison of the temperature distribution under different eccentricity ratios. (**a**) *ε* = 0.3; (**b**) *ε* = 0.5; (**c**) *ε* =0.7; (**d**) maximum temperature.

**Figure 9 micromachines-12-01208-f009:**
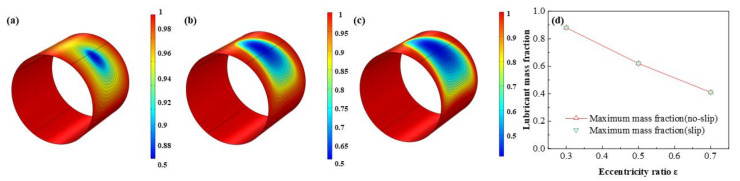
Comparison of the liquid mass fraction distribution under different eccentricity ratios. (**a**) *ε* = 0.3; (**b**) *ε* = 0.5; (**c**) ε = 0.7; (**d**) minimum liquid mass fraction.

**Figure 10 micromachines-12-01208-f010:**
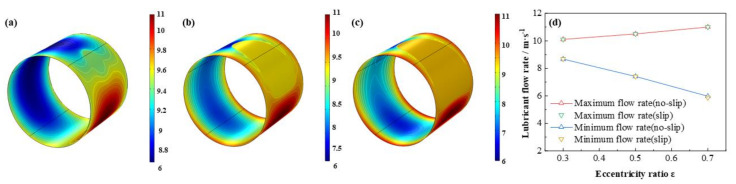
Comparison of the flow rate distribution under different eccentricity ratios. (**a**) *ε* = 0.3; (**b**) *ε* = 0.5; (**c**) *ε* = 0.7; (**d**) maximum/minimum flow rate.

**Figure 11 micromachines-12-01208-f011:**
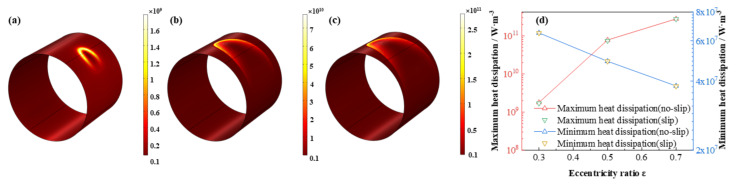
Comparison of the heat dissipation distribution under different eccentricity ratios. (**a**) *ε* = 0.3; (**b**) *ε* = 0.5; (**c**) *ε* = 0.7; (**d**) maximum/minimum heat dissipation.

**Figure 12 micromachines-12-01208-f012:**
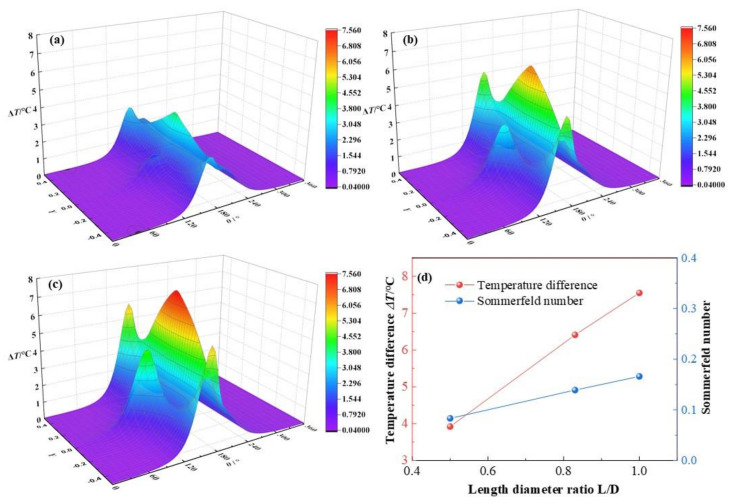
Comparison of the temperature distribution under different length diameter ratios. (**a**) *L*/*D* = 0.5; (**b**) *L*/*D* = 0.83; (**c**) *L*/*D* = 1.0; (**d**) maximum temperature.

**Figure 13 micromachines-12-01208-f013:**
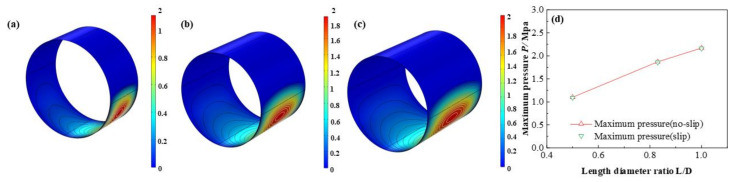
Comparison of the pressure distribution under different length diameter ratios. (**a**) *L*/*D* = 0.5; (**b**) *L*/*D* = 0.83; (**c**) *L*/*D* = 1.0; (**d**) minimum pressure.

**Figure 14 micromachines-12-01208-f014:**
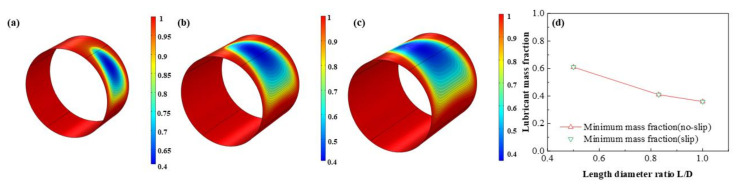
Comparison of the liquid mass fraction distribution under different length diameter ratios. (**a**) *L*/*D* = 0.5; (**b**) *L*/*D* = 0.83; (**c**) *L*/*D* = 1.0; (**d**) minimum liquid mass fraction.

**Figure 15 micromachines-12-01208-f015:**
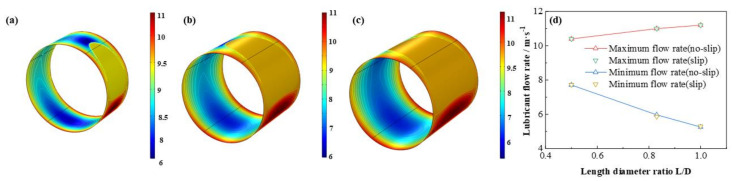
Comparison of the flow rate distribution under different length diameter ratios. (**a**) *L*/*D* = 0.5; (**b**) *L*/*D* = 0.83; (**c**) *L*/*D* = 1.0; (**d**) maximum/minimum flow rate.

**Figure 16 micromachines-12-01208-f016:**
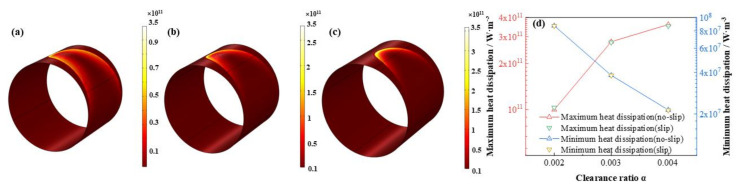
Comparison of the heat dissipation distribution under different length diameter ratios. (**a**) *L*/*D* = 0.5; (**b**) *L*/*D* = 0.83; (**c**) *L*/*D* = 1.0; (**d**) maximum/minimum heat dissipation.

**Figure 17 micromachines-12-01208-f017:**
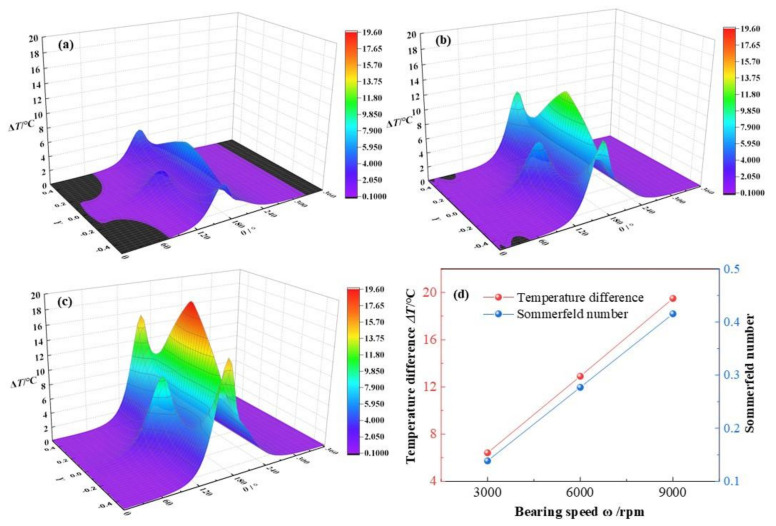
Comparison of the temperature distribution under different bearing speeds. (**a**) *ω* = 3 × 10^3^ rpm; (**b**) *ω* = 6 × 10^3^ rpm; (**c**) *ω* = 9 × 10^3^ rpm; (**d**) maximum temperature.

**Figure 18 micromachines-12-01208-f018:**
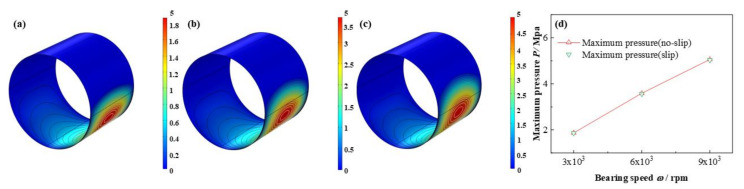
Comparison of the pressure distribution under different bearing speeds. (**a**) *ω* = 3 × 10^3^ rpm; (**b**) *ω* = 6 × 10^3^ rpm; (**c**) *ω* =9 × 10^3^ rpm; (**d**) minimum pressure.

**Figure 19 micromachines-12-01208-f019:**
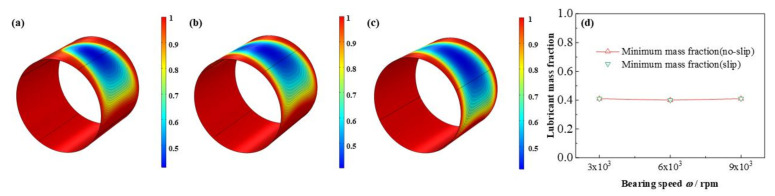
Comparison of the liquid mass fraction distribution under different bearing speeds. (**a**) *ω* = 3 × 10^3^ rpm; (**b**) *ω* = 6 × 10^3^ rpm; (**c**) *ω* = 9 × 10^3^ rpm; (**d**) minimum liquid mass fraction.

**Figure 20 micromachines-12-01208-f020:**
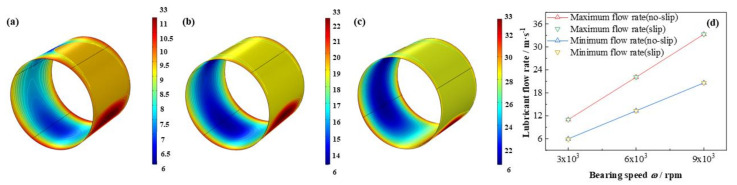
Comparison of the flow rate distribution under different bearing speeds. (**a**) *ω* = 3 × 10^3^ rpm; (**b**) *ω* = 6 × 10^3^ rpm; (**c**) *ω* = 9 × 10^3^ rpm; (**d**) maximum/minimum flow rate.

**Figure 21 micromachines-12-01208-f021:**
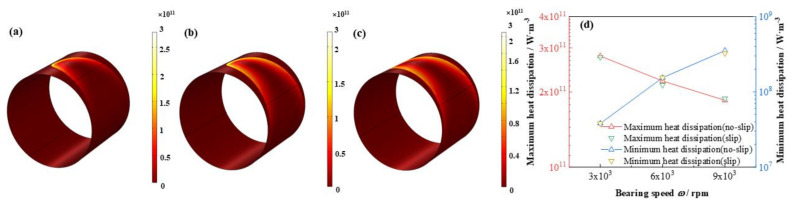
Comparison of the heat dissipation distribution under different bearing speeds. (**a**) *ω* = 3 × 10^3^ rpm; (**b**) *ω* = 6 × 10^3^ rpm; (**c**) *ω* = 9 × 10^3^ rpm; (**d**) maximum/minimum heat dissipation.

**Figure 22 micromachines-12-01208-f022:**
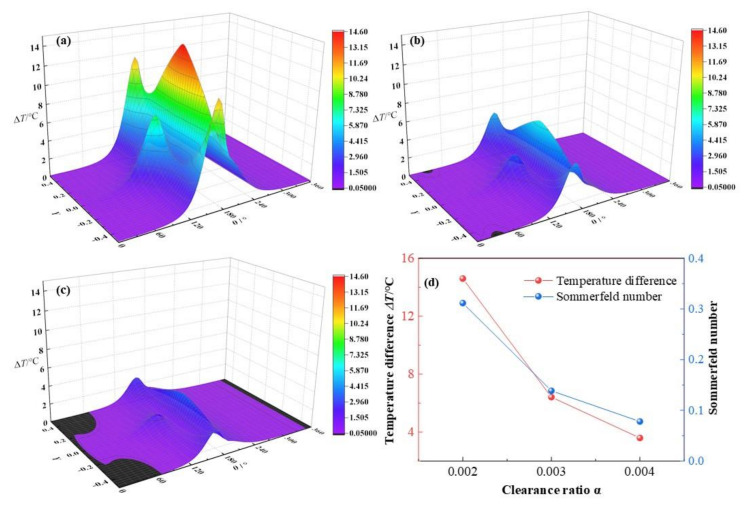
Comparison of the temperature distribution under different clearance ratios. (**a**) *α* = 0.002; (**b**) *α* = 0.003; (**c**) *α* = 0.004; (**d**) maximum temperature.

**Figure 23 micromachines-12-01208-f023:**
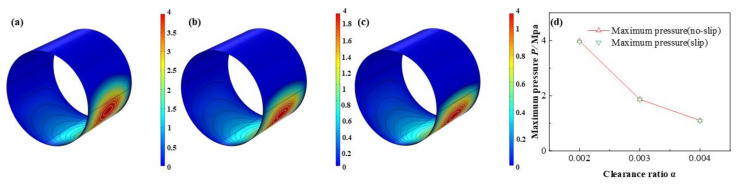
Comparison of the pressure distribution under different clearance ratios. (**a**) *α* = 0.002; (**b**) *α* = 0.003; (**c**) *α* = 0.004; (**d**) minimum pressure.

**Figure 24 micromachines-12-01208-f024:**
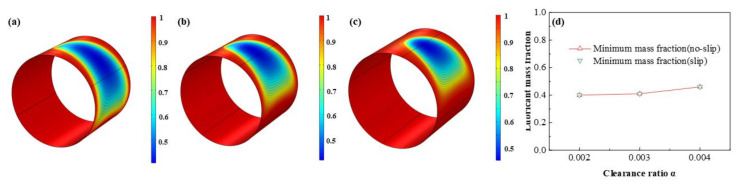
Comparison of the liquid mass fraction distribution under different clearance ratios. (**a**) *α* = 0.002; (**b**) *α* = 0.003; (**c**) *α* = 0.004; (**d**) minimum liquid mass fraction.

**Figure 25 micromachines-12-01208-f025:**
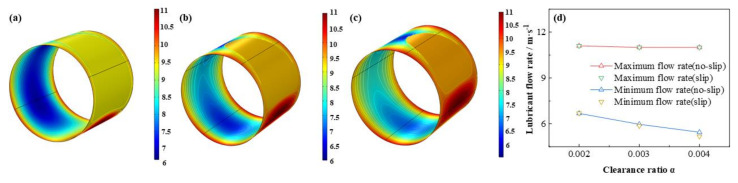
Comparison of the flow rate distribution under different clearance ratios. (**a**) *α* = 0.002; (**b**) *α* = 0.003; (**c**) *α* = 0.004; (**d**) maximum/minimum flow rate.

**Figure 26 micromachines-12-01208-f026:**
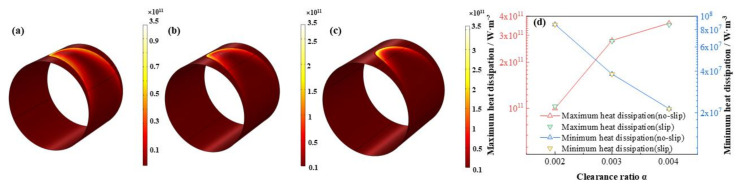
Comparison of the heat dissipation distribution under different clearance ratios. (**a**) *α* = 0.002; (**b**) *α* = 0.003; (**c**) *α* = 0.004; (**d**) maximum/minimum heat dissipation.

**Table 1 micromachines-12-01208-t001:** Bearing geometric and operating parameters.

Parameter Name	Value
Bearing length (*L*), m	6 × 10^−2^~1.2 × 10^−1^
Shaft radius (*R*), m	6 × 10^−2^
Initial Radial clearance (*C*_0_), m	1.2 × 10^−2^~2.4 × 10^−2^
Initial temperature (*T*_0_), K	313
Initial dynamic viscosity of lubricant oil at 313 K (*η*_0_), Pa·s	2.77 × 10^−2^
Initial density of lubricant at 313 K (*ρ*_0_), kg·m^−3^	860
Specific heat capacity of lubricant (*C*), J·(kg·K)^−1^	2000
Heat transfer coefficient of lubricant (*h*), W·(m^2^·K)^−1^	80
Eccentricity ratio (*ε*)	0.3~0.7
Bearing speed (*ω*), rpm	3 × 10^3^~9 × 10^3^

**Table 2 micromachines-12-01208-t002:** Calculation parameters.

Bearing Length (*L*), m	Shaft Radius (*R*), m	Initial Radial Clearance (*C*_0_), m	Eccentricity Ratio (*ε*)	Bearing Speed (*ω*), rpm
0.1 × 10^−1^	0.6 × 10^−1^	0.18 × 10^−3^	0.6	6 × 10^3^

**Table 3 micromachines-12-01208-t003:** Calculation parameters.

Bearing Length (*L*), m	Shaft Radius (*R*), m	Initial Radial Clearance (*C*_0_), m	Lubricant Viscosity at 313 K (*μ*), Pa·s	Bearing Speed (*ω*), rpm
0.8 × 10^−1^	0.5 × 10^−1^	0.145 × 10^−3^	0.0277	6 × 10^3^

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
