# Peer review of "Prediction on Flow and Thermal Characteristics of Ultrathin Lubricant Film of Hydrodynamic Journal Bearing"

_micromachines, 2021, doi:10.3390/mi12101208_

Round 1
Reviewer 1 Report
The manuscript invesitigates the flow and thermal characteristics of a lubricant film in the micro clearance with high rotating speed of hydrodynamically lubricated journal bearings. Numerical modeling was done by means of the Reynolds differntial equation coupled with the energy and viscosity-temperature equations. Cavitation was considered in a mass-conserving way. The authors present 3D surface diagrams of the lubricant film thickness, pressure, temperature, liquid mass fraction, flow rate and heat dissipation distributions under different geometric and operating conditions. The authors report that with the rise of eccentricity or length diameter ratio, the maximum peaks of pressure, temperature and heat dissipation are rapidly increased, the cavitation is aggravated, and the flow rate is accelerated. As the bearing speed accelerated, the maximum peak of temperature increased strongly, whereas, the distinction between peaks of flow rate and heat dissipation is magnified and reduced, respectively.
Generally, numerical modeling is well done and the methods and results are well presented. Therefore, the manuscript could be considered for publication after the following issues were addressed:
- Thematically, lubricant film formation in HL Journal Bearings is nothing new, and given the large amount of literature available, the findings do not seem overly far-reaching (even if they are nicely presented). The authors should definitely elaborate more on the novelty factor of this paper.
- Shear rate dependent (non-Newtonian) effects on the viscosity were neglected. Considering the high sliding velocities, an influence may occur. The authors should discuss this or better even take it into account in the model.
- How was the temperature influence on the density taken into account?
- To take into account the exact temperature distribution in the lubrication gap and thus the influence on the viscosity, a resolution in the direction of the lubrication gap height is actually required, as well as an extended Reynolds equation (e.g. according to Yang and Wen, 10.1007/BF02486885). Why was this omitted here and the simple Reynolds equation used?
- The labeling in Fig. 6-26 is too small and must be enlarged.
- A lot of colorful images can't conceal the not-quite-clear degree of novelty.
Author Response
Please see the attachment.
Special thanks to you for your good comments.

Reviewer 2 Report
The investigation presented in the introduction is a formal one (based on expressions such as: the authors presented / investigated / analyzed / measured / etc.), without being critical / comparative /, highlighting advantages / disadvantages of the mentioned methods / techniques.
The lumped references must be avoided. Include a critical comment for each reference (or two references with the same objectives)
1) Rewrite the introduction.
The research objective is defined.
Mention the novelty at the end of introduction in points-wise manner.
The model and operation conditions are presented.
The model is validated and the main results are presented and discussed.
2) Enlarge the figures for better visibility (using left indentation = 0).
3) include summary tables of the results presented.
The main findings of the research study are presented in conclusion section.
4) Mention the next work based on the results presented in this paper
5) include more reference published in the last 3 years to highlight the topicality of this research study.
Author Response

(The authors gave the same response as above.)

Reviewer 3 Report
The optimization of the finite length journal bearings is of interest to researchers and engineers working in the field.
This paper presents an original model, but the article's structure and the presented results have to be improved.
- The title is unsuited and unusual. The paper studies the hydrodynamic journal bearings and not only its micro clearance. I propose to amend this aspect, choosing the right title.
- In the Introduction section, the contribution to the research field of references [23-26] is not mentioned. In fact, what this paper brings new over the existing literature?
- At line 74: there is no interaction between the temperature and the lubricant, but there is an influence of the temperature on the viscosity of the lubricant.
- Which type of lubricant was adopted? The lubricant properties must be fully specified, not only the initial viscosity (mention the viscosity values at two different temperatures, density, and heat transfer coefficient).
- The results should be synthetically presented, using the Sommerfeld number defined for finite-length journal bearings.
- There is no optimization for the dimensional and functional parameters of the HJB, e.g., maximum load-carrying capacity, optimum eccentricity, HJB's length, and minimum developed temperature. A compromise between the dimensional and running parameters to attain the best performances is necessary.
- The Conclusions section must present results regarding the optimization of the HJB.
Author Response

(The authors gave the same response as above.)

Round 2
Reviewer 1 Report
Thank you for addressing my comments in detail. I'm satisfied by the explanations and changes to the manuscript. Therefore, I recommend acceptance of the paper.
Reviewer 2 Report
The authors revised.
Reviewer 3 Report
The authors responded to all the questions and amended the paper as requested. I recommend the publication of this paper.